# Therapeutic concentrations of calcineurin inhibitors do not deregulate glutathione redox balance in human renal proximal tubule cells

Yasaman Ramazani[1], Noël Knops[1,2], Sante Princiero Berlingerio[1], Oyindamola Christiana Adebayo[1], Celien Lismont[3], Dirk J. Kuypers[4], Elena Levtchenko[1,2], Lambert P. van den Heuvel[1,5☯], Marc Fransen[3☯]*

1 Laboratory of Pediatric Nephrology, Department of Growth and Regeneration, University of Leuven, Leuven, Belgium, 2 Department of Pediatric Nephrology and Solid Organ Transplantation, University Hospitals Leuven, Leuven, Belgium, 3 Laboratory of Peroxisome Biology and Intracellular Communication, Department of Cellular and Molecular Medicine, University of Leuven, Leuven, Belgium, 4 Department of Nephrology and Renal Transplantation and Department of Microbiology, Immunology and Transplantation, University of Leuven, Leuven, Belgium, 5 Translational Metabolic Laboratory and Department of Pediatric Nephrology, Radboud University Medical Center, Nijmegen, The Netherlands

☯ These authors contributed equally to this work.
* marc.fransen@kuleuven.be

**Data Availability Statement:** All relevant data are within the manuscript and its Supporting information files.

## Abstract

The calcineurin inhibitors (CNI) cyclosporine A and tacrolimus comprise the basis of immunosuppressive regimes in all solid organ transplantation. However, long-term or high exposure to CNI leads to histological and functional renal damage (CNI-associated nephrotoxicity). In the kidney, proximal tubule cells are the only cells that metabolize CNI and these cells are believed to play a central role in the origin of the toxicity for this class of drugs, although the underlying mechanisms are not clear. Several studies have reported oxidative stress as an important mediator of CNI-associated nephrotoxicity in response to CNI exposure in different available proximal tubule cell models. However, former models often made use of supra-therapeutic levels of tissue drug exposure. In addition, they were not shown to express the relevant enzymes (e.g., CYP3A5) and transporters (e.g., P-glycoprotein) for the metabolism of CNI in human proximal tubule cells. Moreover, the used methods for detecting ROS were potentially prone to false positive results. In this study, we used a novel proximal tubule cell model established from human allograft biopsies that demonstrated functional expression of relevant enzymes and transporters for the disposition of CNI. We exposed these cells to CNI concentrations as found in tissue of stable solid organ transplant recipients with therapeutic blood concentrations. We measured the glutathione redox balance in this cell model by using organelle-targeted variants of roGFP2, a highly sensitive green fluorescent reporter protein that dynamically equilibrates with the glutathione redox couple through the action of endogenous glutaredoxins. Our findings provide evidence that CNI, at concentrations commonly found in allograft biopsies, do not alter the glutathione redox balance in mitochondria, peroxisomes, and the cytosol. However, at supra-therapeutic concentrations, cyclosporine A but not tacrolimus increases the

**Funding:** This work was funded by the grant from the Research Foundation – Flanders [Onderzoeksproject G095315N]. YR was supported by a doctoral fellowship of the Research Foundation – Flanders [grant number 1S24417N, 2017], and CL was supported by a postdoctoral fellowship of the same organization [grant number 1213620N]. EL is funded by Clinical Investigator grant of Research Foundation – Flanders [grant number 1801110N]. Research Foundation – Flanders: Fonds Wetenschappelijk Onderzoek: https://www.fwo.be. The funders had no role in study design, data collection and analysis, decision to publish, or preparation of the manuscript.

**Competing interests:** The authors have declared that no competing interests exist.

ratio of oxidized/reduced glutathione in the mitochondria, suggestive of imbalances in the redox environment.

## Introduction

Cyclosporine A (CsA) and tacrolimus (Tac), two calcineurin inhibitors (CNI), are globally recognized as the cornerstone of immunosuppressive therapy after solid organ transplantation and, despite their different chemical structure, both CNIs exert their immunosuppressive action via inhibition of the T-cell response. In addition, CNI's are used for many other indications outside the realm of solid transplantation such as nephrotic syndrome, inflammatory bowel disease and different rheumatic diseases. However, their chronic use causes renal toxicity, the so-called CNI-associated nephrotoxicity (CNIT), manifested by interstitial fibrosis, afferent arteriolar hyalinosis, and the loss of renal function [1, 2]. The mechanisms of CNIT are largely unknown, but it has been reported that systemic exposure to CNI increases the formation of free radicals (e.g., superoxide) and other reactive oxygen species (ROS) (e.g., $H_2O_2$), most probably due to vasoconstriction-induced hypoxia [3]. At kidney level, proximal tubule cells (PTC) are a metabolically active cell type that are exposed to high drug concentrations due to active and passive take-up from the blood and filtrate and hence are the most prone to drug-induced injury. They are the only renal cells that metabolize CNI and multiple studies have linked common variations in the genes encoding the CYP3A5 enzyme (*CYP3A5*, Entrez Gene ID: 1577) and the P-glycoprotein (P-gp) efflux transporter (*ABCB1*, Entrez Gene ID: 5243), responsible for CNI metabolism in the PTC, with the risk for developing CNIT [4–8].

Redox homeostasis plays an important role in maintaining health of organisms and, at physiological levels, ROS such as $H_2O_2$ often act as crucial mediators of various cellular functions. However, imbalances in ROS can lead to detrimental effects on cellular and organismal health [9]. The type and concentration of produced ROS as well as their location and elimination kinetics are important factors in determining the magnitude of the physiological and pathological effects [10]. Subcellular compartments such as mitochondria, peroxisomes, the endoplasmic reticulum, and membrane-bound NADPH oxidases are major locations of ROS production [11]. Reduced glutathione (GSH) is a tripeptide that, among other functions, supports antioxidant reactions and protects cells from oxidative insults [10]. While in physiological conditions the concentration of GSH is 10 to 100-fold higher than that of oxidized glutathione (GSSG) [12], in pathophysiological states, a series of enzymatic reactions between the endogenous glutaredoxins and GSH can lead to higher amounts of GSSG. Therefore, the ratio of GSH/GSGG can serve as an indicator of redox changes that may lead to oxidative stress [12, 13].

Previous research has shown the involvement of oxidative stress in several human and animal PTC models in relation to CNI exposure, thus implying that tackling the phenomenon of increased ROS might reduce CNIT [14–18]. However, the majority of these studies have used concentrations of CNI that are 100-200-fold higher than concentrations reached in the kidney under clinically targeted blood trough levels, hindering the clinical relevance of these findings. In addition, these models have not been demonstrated to express the relevant enzymes and transporters for the metabolism/efflux of CNI [19]. Moreover, capturing dynamic changes in the glutathione redox couple in PTC in response to the dynamics of intracellular CNI exposure and metabolism necessitates a controlled real-time system. In this work, we have filled the above-mentioned gaps by assessing the glutathione redox balance at the subcellular level in

real time and by using a validated model for studying *in vitro* CNI metabolism. The concentration range of CNI used was derived from data of human liver and kidney tissue homogenates obtained during protocol biopsies [20–23]. In addition, we have tested supra-therapeutic concentrations of CNI, as in aforementioned studies, to detect potential differences between our state-of-the-art methodology and methods previously published.

## Materials and methods

### Cell culture

Conditionally immortalized proximal tubule cells (ciPTC) were derived from allograft biopsies and the allograft genotype for *CYP3A5 (rs776746)* and *ABCB1 (rs1045642)* was determined on immortalized cells as described elsewhere [24]. ciPTC with the specific genotype of *CYP3A5 *3/*3* and *ABCB1 3435 TT* were selected for CNI incubations, given their most pronounced profibrotic effects in response to Tac exposure *in vitro* in pilot experiments (data submitted for publication). The concentration range of CNI used was derived from data of human liver and kidney tissue homogenates obtained during protocol biopsies corresponding to 15 and 0.3 μg/mL for CsA and Tac, respectively [20–23]. Cells were cultured at 33˚C in a humidified 5% $CO_2$ incubator in DMEM-HAM's F-12 medium (Biowest) supplemented with insulin-transferrin-selenium (Sigma-Aldrich; 5 ng/mL), hydrocortisone (Sigma-Aldrich; 36 ng/mL), epidermal growth factor (Sigma-Aldrich; 10 ng/mL), triiodothyronine (Sigma-Aldrich; 40 pg/mL), 10% v/v fetal bovine serum, penicillin (Gibco; 100 U/mL), and streptomycin (Gibco; 100 U/mL). For redox imaging, approximately 120,000 ciPTC were seeded into FD35 FluoroDishes (World Precision Instruments). One day later, the FluoroDishes were transferred from a 33˚C to a 37˚C incubator to allow differentiation for 7 days before being transfected. One day post-transfection, the medium was replaced with fresh medium supplemented with tacrolimus (Tac; Merck), cyclosporine A (CsA; Merck), L-buthionine-sulfoximine (BSO; Sigma), and/or dimethyl sulfoxide (DMSO, Sigma). DMSO was maintained at 0.1% v/v in all conditions, except for the 50 μg/mL Tac (0.5% v/v) and BSO control (0% v/v) conditions. Redox measurements were performed 24-h and 48-h post-treatment (after measurement, the media were replaced with fresh medium). 1 mM hydrogen peroxide ($H_2O_2$) in PBS was used as a positive control. For metabolomics analysis, 200,000 ciPTC were seeded in 6-well plates (VWR). One day later, the plates were transferred from a 33˚C to a 37˚C incubator for at least 7 days to allow differentiation. Next, the medium was replaced with fresh medium supplemented with Tac (Merck), CsA (Merck), BSO (Sigma), or 0.1% v/v DMSO.

### Ethics statement

The study protocol was approved by the Institutional Ethical Board (B32220109632 and B322201317188) and written consent was obtained from all study subjects or their parents. The study is registered with the clinical trial number of NCT01767350.

### Plasmid isolation and transfection

Prior to transfection, the plasmids encoding cytosolic roGFP2 (pMF1707) [25], peroxisomal roGFP2 (pMF1706) [25], mitochondrial roGFP2 (pMF1762) [25], cytosolic roGFP2-Orp1 (pMF1834) [26] or mitochondrial roGFP2-Orp1 (pMF1833) [26] were purified using the Pure-Link™ HiPure Plasmid Midiprep Kit (ThermoFisher Scientific) according to the manufacturer's protocol. Briefly, bacterial cultures were pelleted (4,000 x g, 10 min, room temperature) and resuspended in RNAse A-containing resuspension buffer. After addition of lysis buffer, the samples were gently mixed. Next, precipitation buffer was added and the samples were

centrifuged (12,000 x g, 10 min, room temperature). The cleared supernatants were loaded onto pre-equilibrated PureLink™ HiPure Midi Columns. After draining, the columns were washed twice with wash buffer and the plasmid DNA was subsequently eluted with elution buffer. Next, the plasmids were precipitated with isopropanol and pelleted by centrifugation (12,000 x g, 30 min, 4˚C). Finally, the pellets were resuspended in 70% v/v ethanol, re-pelletted by centrifugation (12,000 x g, 5 min, 4˚C), air-dried, and resuspended in TE buffer. The purified plasmids were used to transfect ciPTC using the JetPRIME[R] reagent (PolyPlus-transfection). Therefore, 2 μg of each purified plasmid was mixed with 200 μL of Jetprime buffer, vortexed for 10 seconds, and spun down. Next, 4 μL of the JetPRIME reagent was added, and the mixture was vortexed and briefly spun down again. Finally, after incubation for 10 min at room temperature, the mixture was added to ciPTC pre-seeded in FluoroDishes and cultured in serum-containing medium.

### Fluorescence microscopy and redox imaging

Fluorescence was evaluated by using a previously described microscopic setup [27] or by using a motorized inverted IX-81 microscope, controlled by cellSens Dimension software (version 2) and equipped with (i) a temperature, humidity, and $CO_2$-controlled incubation chamber, (ii) a CoolLED pE-4000 illumination system, (iii) a 100x Super Apochromat oil immersion objective, (iv) the band-pass excitation filters D405/20x and BP470-495 (v) the barrier filter BA510-550 (vi) a dichromatic mirror with a cut-off 505 nm, and (vii) a DP73 high-performance Peltier-cooled digital color camera (Olympus Belgium). Ratiometric measurements of roGFP2 or roGFP2-Orp1 oxidation were performed as described previously [28]. Briefly, the camera exposure times were set to a ratio of 1:2 to acquire images at the 400 (for GSSG) and 480 (for GSH) nm excitation wavelengths, respectively. Random fields of cells were imaged with the same 400/480 nm exposure time ratios, and images having signal intensities within the linear dynamic range of the camera were retained for analysis. The cellSens image analysis software (Olympus Belgium) was used to quantify the relative fluorescence intensities of roGFP2 or roGFP2-Orp1 (recorded around 510 nm) at 400 nm and 480 nm excitation. For the background subtraction, regions-of-interest were selected on neighboring cell-free areas. The 400/480 nm fluorescence response ratios of the GSH/GSSG sensor were measured and normalized to the average value of the corresponding vehicle values, which were set to 100 arbitrary units (AU).

### Metabolomic analysis of glutathione species

Metabolite extraction was performed essentially as described elsewhere [29]. Briefly, the cells were washed once with ice cold 0.9% w/v NaCl solution, and metabolite extraction was started by addition of 80% v/v methanol and 0.2% w/v of myristic acid-d27 (internal standard). Five minutes later, the cells were scraped, collected in a new tube, and centrifuged at 20,000 x g for 10 min (at 4˚C). The supernatants were transferred to a new vial for mass spectrometry (MS) analysis, and the pellets were used for protein quantification using the Pierce bicinchoninic acid assay kit (ThermoFisher). Ten μL of each sample was injected into a Dionex UltiMate 3000 liquid chromatography system (ThermoFisher Scientific) equipped with a C18 column (Acquity UPLC -HSS T3 1. 8 μm; 2.1 x 150 mm, Waters) coupled to a Q Exactive Orbitrap mass spectrometer (ThermoFisher Scientific) operating in negative ion mode. A step gradient was carried out using solvent A (10 mM tributylamine and 15 mM acetic acid) and solvent B (100% methanol). The gradient started with 0% of solvent B and 100% solvent A and remained at 0% B until 2 min post-injection. A linear gradient to 37% B was carried out until 7 min, and then the concentration of solvent B was linearly increased to 41% from 7 to 14 min. Between 14 and 26 min, the gradient increased to 100% of B and remained at 100% B for 4 min. At 30

min, the gradient returned to 0% B, and chromatography was stopped at 40 min. The flow was kept constant at 250 μL/min at the column was placed at 25˚C throughout the analysis. The MS operated in full scan mode (m/z range: [70–1050]) using a spray voltage of 3.2 kV, capillary temperature of 320˚C, sheath gas at 10.0 bar, auxiliary gas at 5.0 bar. The automatic gain control target was set at 3e6 using a resolution of 140,000, with a maximum ion trap fill time of 512 ms. Data collection was performed using the Xcalibur software (ThermoFisher Scientific). The data analyses were performed by integrating the peak areas using the EL-MAVEN on Polly interface (Elucidata). The raw abundance values of GSSG and GSH were first normalized to the internal standard myristic acid-d27 and subsequently to the total amount of protein. The GSSG/GSH ratios were obtained by dividing the normalized values of GSSG and GSH.

## Cell viability assay

Cell viability was evaluated with the PrestoBlue™ HS cell viability reagent (Invitrogen), a resazurin-based assay that utilizes the mitochondrial metabolic capacity to reduce resazurin, the indicator dye, to resorufin, a strongly fluorescent compound. In brief, ciPTC were trypinized and seeded at a density of 5,000 cells per well in a 96-well black clear-bottom culture plate (Grenier). The cells were first incubated at 33˚C for 8–16 hours to allow attachment, and subsequently transferred to 37˚C for 7–10 days to allow differentiation. Next, the cells were exposed to the test compounds for 48 h (final volume: 100 μL). Thereafter, 10 μL of the PrestoBlue™ HS cell viability reagent was added per well, and the cells were further incubated in the dark at 37˚C for 1 h. Afterwards, resorufin fluorescence was measured using a scanning fluorescence microplate reader (Molecular Devices, Sunnyvale, CA, USA) at excitation/emission wavelengths of 560/590 nm. Three independent experiments were performed with four technical replicates each. The values were normalized to the non-treated cells and presented as the percentage of cell viability.

## Normalization and statistical analysis

Due to unforeseen configuration changes in the live-cell imaging setup during the course of our experiments, all 400/480 nm ratiometric responses were normalized to the average of the corresponding control condition (DMSO or basal), which was set to 100 arbitrary units (AU). Every graph represents the combined data of 3 independent experiments, unless stated otherwise. The confidence interval was set at 95% for all analyses and the null hypothesis was rejected with a p-value <0.05.

Shapiro-Wilk test was used for the assessment of normality. One-way ANOVA was used for detecting the differences between more than two groups for normally distributed data and Kruskal-Wallis for non-normally distributed data. Two-tailed paired one-sample t-test and Wilcoxon test were used for detecting the differences of two groups for normal and non-normally distributed data, respectively. One-tailed unpaired one-sample t-test with Welch correction was used for the differences between the percentage of cells with mitochondrial fragmentation. PRISM version 9 was used for statistical analyses.

## Results

### Exposure of ciPTC to therapeutic concentrations of CsA or Tac does not induce significant changes in the mitochondrial, peroxisomal, or cytosolic glutathione redox balance

To monitor potential changes in the subcellular glutathione redox balance in ciPTC exposed to therapeutic concentrations of CNI found in kidney tissue *in vivo* (i.e. 15 μg/mL of CsA and

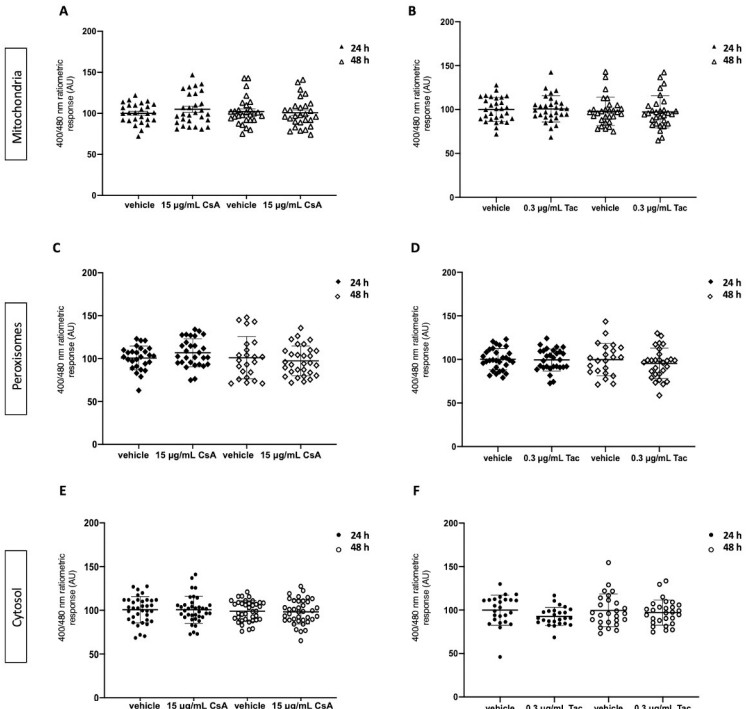

**Fig 1. Mitochondrial, peroxisomal, and cytosolic glutathione redox ratios in ciPTC exposed to therapeutic tissue concentrations of CNI for 24 and 48 h.** CiPTC expressing mitochondrial, peroxisomal, or cytosolic roGFP2 were exposed to (**A**, **C**, **E**) vehicle (0.1% v/v DMSO) or 15 μg/mL of cyclosporin A (CsA) and (**B**, **D**, **F**) vehicle (0.1% v/v DMSO) or 0.3 μg/mL tacrolimus (Tac). Each graph summarizes the results of 3 independent experiments, depicting the median and standard deviation (as error bars). Every individual symbol represents the average ratio of 10 measurements within one cell (at least 8 randomly chosen cells were analyzed per experiment). Kruskal-Wallis (KW) or One-way ANOVA (OWA) test were used to calculate p-values: (**A**) 0.8 (KW); (**B**) 0.4 (KW); (**C**) 0.2 (OWA); (**D**) 0.6 (OWA); (**E**) 0.8 (OWA); and (**F**) 0.2 (KW).

0.3 μg/mL of Tac), we employed targeted variants of roGFP2, a highly responsive GSH/GSSG sensor. As depicted in Fig 1, exposure of ciPTC to the above-mentioned concentrations of CsA or Tac did not significantly alter the glutathione redox balance in mitochondria (panels A and B), peroxisomes (panels C and D), or the cytosol (panels E and F), neither at 24 h nor at 48 h of incubation (the mean values of each condition, expressed as arbitrary units and compared to the corresponding control group, are listed in S1 Table). To confirm the responsiveness of roGFP2 to changes in the intracellular glutathione redox balance, we exposed ciPTC expressing cytosolic or mitochondrial roGFP2 to 1 mM external $H_2O_2$ for 15 min. As shown in S1 Fig and S1 Table, such treatment resulted in a significant increase in the GSSG/GSH ratio, both in the cytosolic and mitochondrial compartments.

## Exposure of ciPTC to therapeutic concentrations of CsA or Tac does not induce significant changes in the global glutathione redox balance

To confirm and extend our roGFP2-based measurements, we also carried out a comparative metabolic analysis of GSH and GSSG in ciPTC with or without therapeutic concentrations of CsA or Tac. As a positive control for glutathione depletion, we included a condition in which the cells were treated with 0.5 mM M-buthionine-S,R-sulfoximine (BSO), a potent glutathione (GSH)-depleting agent that inhibits γ-glutamylcysteine synthetase [30]. From these experiments, it is clear that therapeutic CNI concentrations do not induce alterations in the

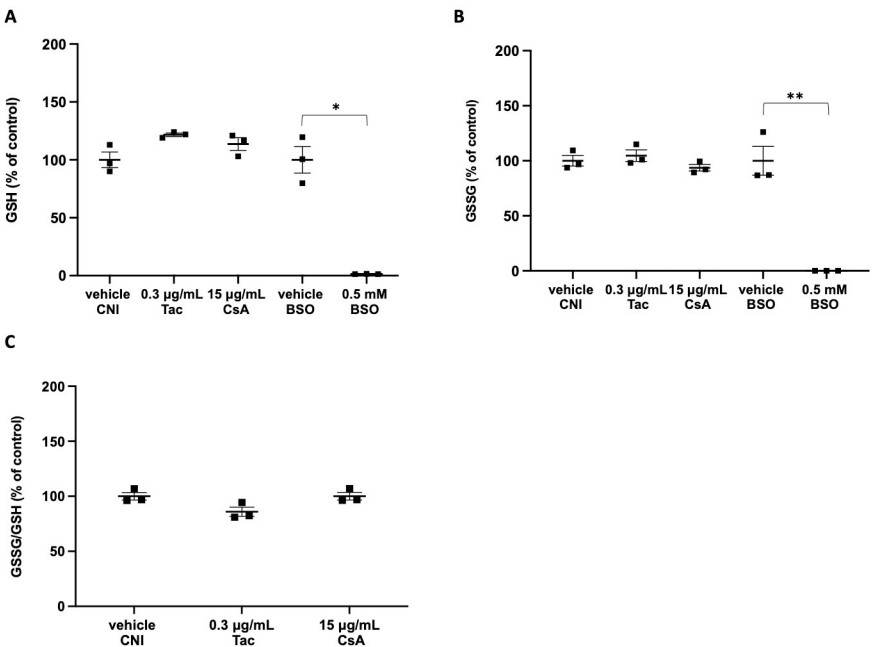

**Fig 2. Metabolomic analysis of glutathione species in ciPTC exposed to CNI or BSO.** CiPTC were exposed for 48 h to therapeutic concentrations of CNI (15 μg/mL of cyclosporin A (CsA), 0.3 μg/mL of tacrolimus (Tac), vehicle (0.1% v/v DMSO) or 0.5 mM L-buthionine-sulfoximine (BSO). Changes in the relative levels of **(A)** GSH, **(B)** GSSG, and **(C)** GSSG/GSH ratios, all expressed as percentages of the average of the corresponding control samples, were determined as described in the Materials and Methods section. Given that, in the 0.5 mM BSO condition, the GSSG levels were below the detection limit, no GSSG/GSH ratio could be calculated. Each graph depicts the mean and standard error (as error bars) of 3 replicates (indicated as filled squares) within one experiment. Data of the control (vehicle) and treated conditions were statistically compared using a paired t-test, and statistical differences are marked (*, $p < 0.05$; **, $p < 0.01$).

glutathione redox balance, neither at the GSH or GSSG level nor in the GSSG/GSH ratio (Fig 2). However, treatment of ciPTC with 0.5 mM BSO decreased the intracellular glutathione levels dramatically within 48 h (Fig 2), a finding in agreement with a previous study [31]. In this context, it has to be mentioned that we also investigated the responsiveness of cytosolic roGFP2 to BSO, both in the absence or presence of therapeutic CNI concentrations (S2 Fig). Importantly, BSO in itself did not cause any significant changes in the oxidation state of the cytosolic glutathione redox sensor. In addition, there were no synergistic effects between BSO and CNI. At first sight, the lack of responsiveness of cytosolic roGFP2 to BSO-induced glutathione depletion may be surprising. However, similar findings have been reported by others [31]. In addition, it can be expected that the relationship between GSH deficiency and subcellular oxidation may vary depending on cell type [25, 32, 33], the subcellular localization of the sensor [31], and the dose and duration of BSO treatment [34].

## Exposure of control or glutathione-depleted ciPTC to therapeutic CNI concentrations does not induce significant changes in cell viability

Given that glutathione depletion is widely used to sensitize cells to oxidative insults, we next investigated whether or not treatment of ciPTC with BSO sensitized these cells to the potential cytotoxic effects of CNI. Interestingly, neither treatment with 0.5 mM BSO nor therapeutic or supra-therapeutic doses (166-fold higher than the therapeutic tissue concentrations) of Tac, alone or in combination, reduced cell viability significantly (Fig 3A). In contrast,

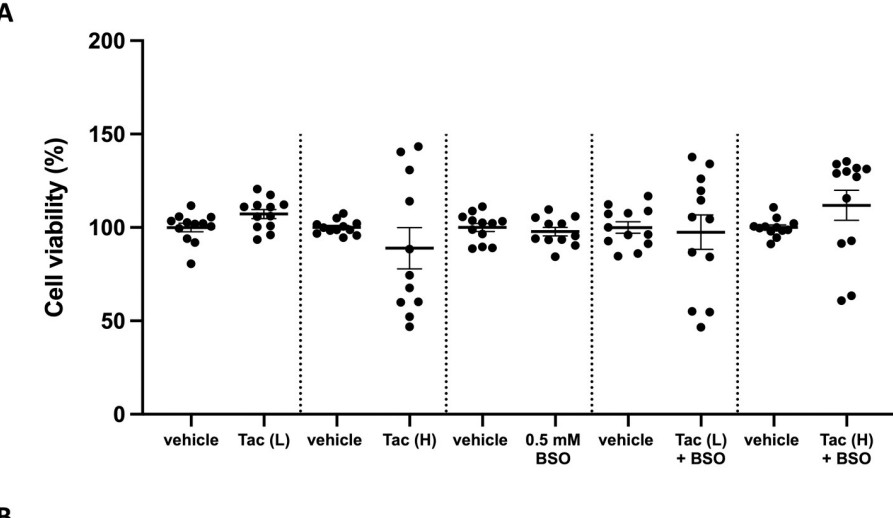

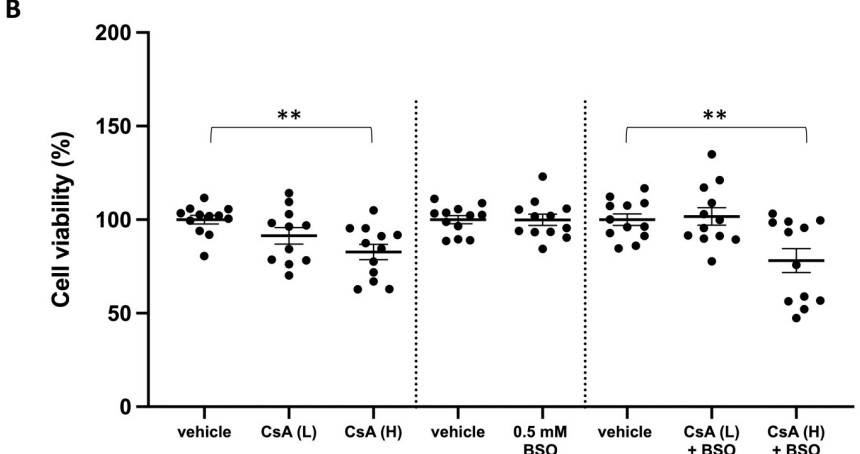

**Fig 3. Cell viability in glutathione-depleted ciPTC exposed to CNI.** CiPTC were exposed for 48 h to **(A)** 0.3 μg/mL of tacrolimus (Tac (L); vehicle: 0.1% v/v DMSO), 50 μg/mL Tac (Tac (H); vehicle: 0.5% DMSO), 0.5 mM L-buthionine-sulfoximine (BSO; vehicle: basal medium), or a combination of each Tac concentration with BSO (vehicle conditions: 0.1% and 0.5% DMSO, respectively); or **(B)** 15 μg/mL cyclosporin A (CsA (L); vehicle: 0.1% v/v DMSO), 50 μg/mL CsA (CsA (H); vehicle: 0.1% v/v DMSO), 0.5 mM BSO (vehicle: basal medium) or a combination of each CsA concentration with BSO (vehicle conditions: 0.1% v/v DMSO). Resorufin fluorescence values for every condition are expressed as percentages of the average of the corresponding control samples as described in the Materials and Methods section. Each graph represents the mean of 3 independent experiments. The large and small horizontal lines show the median and standard errors, respectively. Paired t-test was used to detect significant reduction in cell viability. *, p-value < 0.05; **, p-value < 0.01.

supra-therapeutic concentrations (3.3-fold higher than the therapeutic tissue concentrations) of CsA promoted cell death, both in control and glutathione-depleted cells (Fig 3B). In summary, these findings strongly indicate that therapeutic CNI tissue concentrations do not cause redox stress in GSH-depleted ciPTC, at least not in the time window considered here.

## Exposure of ciPTC to supra-therapeutic concentrations of CsA, but not Tac, results in a small but significant increase in the mitochondrial redox balance

To put our results in perspective, we compared our approach with that of other recent studies in human PTC [17, 18], and monitored how supra-therapeutic concentrations (50 μg/mL) of

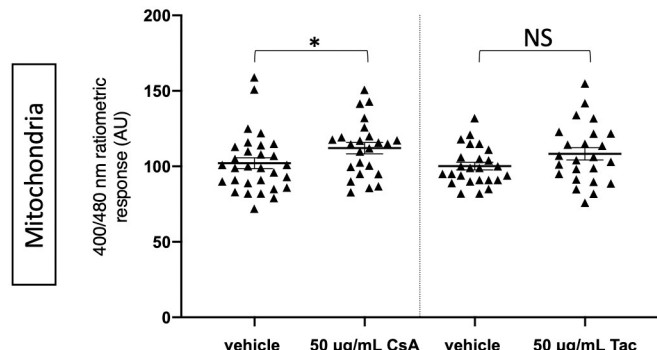

**Fig 4. Mitochondrial glutathione redox ratios in ciPTC exposed to supra-therapeutic concentrations of CNI.**
CiPTC expressing mitochondrial roGFP2 were exposed for 24 h to 50 μg/mL CsA, 50 μg/mL Tac, or vehicle (for CsA: 0.1% v/v DMSO; for Tac: 0.5% v/v DMSO). The results are derived from 3 independent experiments, depicting the median and standard deviation (as error bars). Every individual symbol represents the average ratio of 10 measurements within one cell (at least 8 randomly chosen cells were analyzed per experiment). The Wilcoxon test (WT) and paired t-test (PTT) were used to calculate p-values: *, 0.02 (WT); non-significant (NS), 0.2 (PTT).

CsA and Tac, which were respectively 3.3- and 166-fold higher than the maximum therapeutic tissue concentrations, affected the mitochondrial glutathione redox balance. These experiments demonstrated a small (1.1-fold) but significant increase in the GSSG/GSH ratio at 50 μg/mL CsA compared to vehicle. Exposure to the same concentration of Tac, however, did not cause such a significant change (Fig 4 and S1 Table). Note that these observations are in line with the data presented in Fig 3.

During our experiments, we noticed that conditions in which ciPTC were treated with higher concentrations of DMSO or supra-therapeutic CNI resulted in enhanced mitochondrial fragmentation, even after 24 h of treatment (Fig 5, compare panels A and B). Further analysis demonstrated an approximately 2-fold increase in the percentage of cells with fragmented mitochondria in 0.5 vs. 0.1% v/v DMSO (Fig 5C). In addition, supra-therapeutic concentrations of both CNI in comparison to their vehicle counterparts yielded a 2-3-fold increase in the percentage of cells with mitochondrial fragmentation (Fig 5C). The numerical data are provided in S2 Table.

## Discussion

CNIs are an important class of immunosuppressive medication but are unfortunately associated with nephrotoxicity, of which the underlying pathophysiological mechanism is not well understood [1, 2]. Earlier studies have suggested that CNIT can be linked to oxidative stress in PTC, an observation that may offer new perspectives to reduce the burden of CNIT by antioxidant treatment. However, these studies were hampered due to the unavailability of a suitable renal cell model and the use of supra-therapeutic CNI exposure (for examples, see S3 Table) to mimic the presence of oxidative stress in relation to the *in vivo* situation. Our study is distinctive with respect to the ability of our PTC model to metabolize CNIs and due to the use of CNI concentrations obtained from allograft tissues, which will lead to totally different levels of intracellular exposure to the parent CNI and their metabolites [24]. Within this context, we have measured the glutathione redox balance in mitochondria, peroxisomes, and the cytosol of donor-derived ciPTC exposed to both therapeutic and supra-therapeutic concentrations of CNI. Apart from the validated cell-model, the novelty of this study lies in the use of organelle-targeted variants of roGFP2, a ratiometric fluorescent sensor for measuring changes in the glutathione redox balance in real-time in a setting controlled for temperature and $CO_2$. The

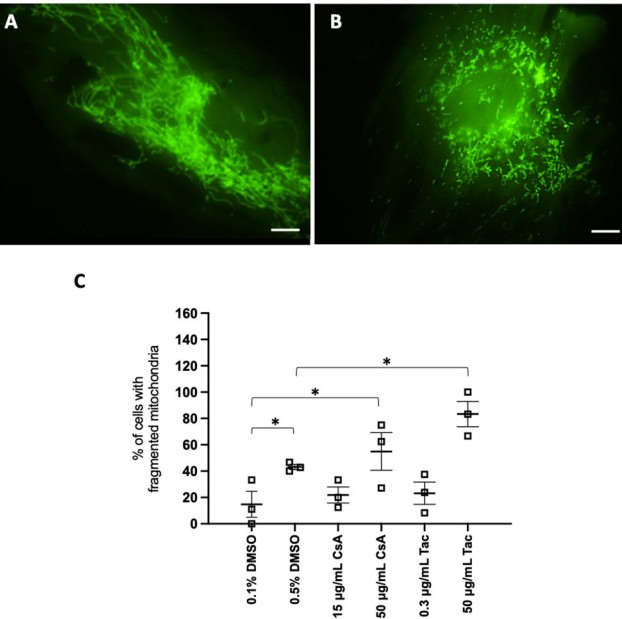

**Fig 5. Mitochondrial morphology in ciPTC exposed to DMSO and CNI.** ciPTC expressing mitochondrial roGFP2 were exposed for 24 h to 0.1% v/v DMSO (vehicle for CsA and 0.3 µg/mL Tac), 0.5% v/v DMSO (vehicle for 50 µg/mL Tac), 15 µg/mL CsA, 50 µg/mL CsA, 0.3 µg/mL Tac, or 50 µg/ml Tac. Representative images of ciPTC with (**A**) elongated or (**B**) fragmented mitochondria. Scale bar, 10 µm. (**C**) Percentage of cells with fragmented mitochondria. *, p-value $< 0.05$ (unpaired t-test with Welch correction), n: 27-39-45-23-41-29. Exact p-values are listed in S2 Table. The results are depicted as the mean and standard error (error bars). Every symbol represents the percentage of cells from one independent experiment (n = 3).

radiometric measurement is particularly interesting, as it eliminates the variations caused by photobleaching, sensor concentration, variable cell thickness, and nonuniform sensor distribution [28]. In addition, we have monitored the GSH/GSSG changes.

In contrast with previous studies, this work demonstrates that exposure to CsA or Tac at therapeutic concentrations does not alter the mitochondrial, peroxisomal, and cytosolic glutathione redox balance in ciPTC after 24 and 48 h. However, treatment of these cells with a supra-therapeutic concentration of CsA, but not Tac, did increase the glutathione redox state in mitochondria, a phenomenon indicative of induction of oxidative stress. Similar findings were observed when targeted variants of roGFP2-Orp1, a highly responsive $H_2O_2$ sensor, was used instead of roGFP2 (S3 and S4 Figs, S4 Table). However, given that (i) this sensor also responds to changes in the glutathione redox balance, and (ii) the responses of roGFP2 and roGFP2-Orp1 are similar in ciPTC exposed to therapeutic (compare Fig 1 and S3 Fig) and supra-therapeutic (compare Fig 4 and S4 Fig) CNI concentrations, these findings should be interpreted with caution and rather be considered as independent replicas.

Exposure of ciPTC to higher concentrations of vehicle (0.5% v/v DMSO) or supra-therapeutic CNI concentrations resulted in a significantly higher percentage of cells with fragmented mitochondria, a phenomenon that can potentially lead to excess ROS production and cytochrome c-dependent apoptosis [35]. These morphological observations provide another evidence on the importance of performing *in vitro* studies within therapeutic CNI ranges that also enable the use of less than 0.5% v/v DMSO. The findings in this study are in line with clinical studies demonstrating that, although the act of solid organ transplantation can transiently induce oxidative stress in the allograft tissue through ischemia-reperfusion injury in the first

period after transplantation [36], this condition can be corrected by the antioxidant systems and improvements in lipoprotein metabolism [36–38].

Our data regarding the effect of a therapeutic concentration of CsA contradict two previous studies in human PTC that reported CsA-induced increase of ROS as a mediator of apoptosis and nephrotoxicity at an even lower concentration (8 μg/mL) [14, 16]. One possible explanation for such contradiction could be the validated capacity for CNI metabolism in our PTC model which can result in alternate intracellular levels of the parent compound or its metabolites responsible for ROS generation. The toxic effects of Tac metabolites are still not established, however, several metabolites of CsA are known to be nephrotoxic [39–41]. Another explanation could be the use of non-reversible (chemi)florescent probes for detection of ROS in these studies. Despite the common use of these chemical probes, they are highly light-sensitive and their irradiation by light can lead to irreversible artefacts in imaging and overestimation of ROS [42, 43]. Such overestimation could also be an explanation for increased ROS at 50 μg/mL Tac in other papers [15, 17, 44], contrary to stable glutathione redox balance with the same concentration of Tac using the highly sensitive roGFP2 in this work. In this context, it is relevant to point out that some researchers use fusion constructs between roGFP2 and glutaredoxin (Grx) to promote rapid and specific equilibration between the dithiol-disulfide pairs of roGFP2 and the local glutathione pool. However, given that (i) others have demonstrated that, inside cells, roGFP2 by itself can already rapidly equilibrate with the local glutathione pool [45], (ii) we have previously shown that all roGFP2 reporters used in this study can respond quickly and reversibly to local changes in redox state in response to transient oxidative insults and intermittent starvation [25] and (iii) in this study, the time window between CNI treatment and roGFP2 measurements is one or two days, a period sufficiently long to ensure equilibration between the roGFP2 sensors and the local glutathione redox couple, it is safe to state that our approach yields reliable results. This claim is further strengthened by our comparative metabolic analysis of GSH and GSSG in ciPTC treated or not with therapeutic CNI concentrations (Fig 2) as well as our observation that such treatments do not impact cytosolic and mitochondrial $H_2O_2$ levels (S3 Fig).

The different outcomes of the previous studies and these investigations can also lie in the kinetics of ROS measurement. We selected the time points based on the previous studies that had observed induction of oxidative stress in PTC as early as 24 h [14, 16]. It remains speculative how these time points would correlate with the chronic exposure of CNI after transplantation. Nonetheless, our selected kinetics are in accordance with the findings on the increase of the CNI-induced toxicity markers after 2–3 days in ciPTC (data submitted for publication). The importance of the time of measurement is evident from *in vivo* studies, in which daily consumption of CNI for longer periods increases the markers of oxidative stress in the blood or urine of animals [40, 46–48]. However, besides a direct effect of CNI on the PTC, the findings from *in vivo* studies can also reflect the vasoconstrictive effects of CNI in inducing ischemia and hypoxia [3]. This hypoxic effect in the interplay between the PTC and endothelial cells is absent in our study and other *in vitro* studies. However, although animal studies could be important in unraveling the mechanisms of kidney toxicity by CNI, the idiosyncratic differences in the expression of relevant actors of intracellular metabolism between species and the translation of relevant drug dosages from the human situation makes their interpretation difficult. Furthermore, clinical studies do not support a major role of oxidative stress in patients after transplantation [37, 38].

Our metabolic analysis of ciPTC treated or not with 0.3 μg/ml Tac or 15 μg/ml CsA provide strong evidence that exposure of these cells to therapeutic CNI concentrations do not significantly deregulate the glutathione redox balance, neither at the GSH or GSSG level nor in the GSSG/GSH ratio (Fig 2). However, previous research has demonstrated lower reduced

glutathione levels after exposure of human kidney 2 (HK-2) cells to 10 μM of CsA [16] or pig kidney proximal tubule (LLC-PK$_1$) cells to 50 μM of Tac [49] for 24 h, which can point toward the sensitivity of these cells to oxidative insults. Moreover, CsA and Tac can differentially regulate the expression of P-gp as a protective response. Hauser *et al*. showed that, at 0.08–1.28 μg/mL CsA and 0.5–1 μg/mL Tac, CsA (but not Tac) increases the expression of P-gp in a concentration-dependent manner [50]. This differential regulation can also affect the response of cells to oxidative stimuli [51].

In conclusion, this study shows the importance of selecting the appropriate model and methods for studying drug toxicity and is the *first* to examine the organelle-targeted redox state of PTC in relation to CNI exposure as a potential contributing factor to the development of CNIT. By combining essential determinants of redox hemostasis in an optimal setting, *in casu* (i) a representative donor-derived PTC model, (ii) clinically relevant CNI exposure, and (iii) spatiotemporal detection of the glutathione redox balance in real-time, we demonstrated that oxidative stress is not likely to be involved in CNIT on PTC level.

## Supporting information

**S1 Fig. Mitochondrial and cytosolic glutathione redox state in ciPTC exposed to hydrogen peroxide.** CiPTC expressing mitochondrial or cytosolic roGFP2 were exposed or not to 1 mM H$_2$O$_2$ for 15 min. The large and small horizontal lines show the median and standard deviations, respectively. Every individual symbol represents the average ratio of 10 measurements within one cell (at least 6 randomly chosen cells were analyzed per experiment). The Wilcoxon test was used to calculate p-values, and statistical differences are marked (*, $p < 0.05$; **, $p < 0.01$).
(TIF)

**S2 Fig. Cytosolic glutathione redox state in ciPTC exposed to a combination of therapeutic CNI levels and BSO.** CiPTC expressing cytosolic roGFP2 were exposed for 48 h to vehicle (0.1% v/v DMSO), 15 μg/mL of cyclosporin A (CsA), 0.3 μg/mL tacrolimus (Tac), 0.5 mM L-buthionine-sulfoximine (BSO), or a combination of CsA or Tac with BSO. The large and small horizontal lines show the median and standard deviations, respectively. Every individual symbol represents the average) ratio of 10 measurements within one cell (at least 9 randomly chosen cells were analyzed per experiment; number of independent experiments: 2). Data were statistically analyzed using one-way ANOVA, but no significant differences were detected.
(TIF)

**S3 Fig. Mitochondrial and cytosolic H$_2$O$_2$ levels in ciPTC exposed to therapeutic CNI concentrations for 24 and 48 h.** CiPTC expressing mitochondrial or cytosolic roGFP2-Orp1 were exposed to (**A**, **C**) vehicle (0.1% v/v DMSO) or 15 μg/mL of cyclosporin A (CsA) and (**B**, **D**) vehicle or 0.3 μg/mL tacrolimus (Tac). The upper and lower panels summarize the results of 3 and 1 experiment, respectively. The large and small horizontal lines show the median and standard deviations (as error bars), respectively. Every individual symbol depicts the average ratio of 10 measurements within one cell (at least 5 randomly chosen cells were analyzed per experiment). One-way ANOVA test was used to calculate p-values: (**A**) 0.1; (**B**) 0.1, (**C**) 0.9; and (**D**) 0.1.
(TIF)

**S4 Fig. Mitochondrial H$_2$O$_2$ levels in ciPTC exposed to supra-therapeutic CNI concentrations.** ciPTC expressing mitochondrial roGFP2-Orp1 were exposed for 24 h to vehicle (for CsA: 0.1% v/v DMSO; for Tac: 0.5% v/v DMSO), 50 μg/mL CsA, or 50 μg/ml Tac. The results are derived from 3 independent experiments. The large and small horizontal lines show the

median and standard deviations (as error bars), respectively. Every individual symbol depicts the average ratio of 10 measurements within one cell (at least 5 randomly chosen cells were analyzed per experiment). The paired t-test was used to calculate p-values: *, 0.01; non-significant (NS), 0.6.
(TIF)

**S1 Table. Mean normalized values of the mitochondrial, peroxisomal, and cytosolic roGFP2-based glutathione redox sensors in ciPTC exposed to calcineurin inhibitors or hydrogen peroxide.** For experimental details, see legend to Fig 1. The percentages represent the mean 400/480 nm ratiometric response ratios compared to the averages of the corresponding non-treated control cells, which were set to 100%. Given that peroxisomes have the capacity to resist oxidative insults generated outside the organelle [1], cells expressing peroxisomal roGFP2 (PO) were not challenged with external $H_2O_2$. The Kruskal-Wallis (KW) or One-way ANOVA (OWA) test were used to calculate p-values. C, cytosolic roGFP2; ciPTC, conditionally immortalized proximal tubule cell; CsA, cyclosporin A; MT, mitochondrial roGFP2; Tac, tacrolimus.
(PDF)

**S2 Table. Quantitative analyses of mitochondrial morphology in ciPTC under CNI.**
(PDF)

**S3 Table. Details of the articles studying ROS production in response to CNI in PTC with different origins.**
(PDF)

**S4 Table. Mean normalized values of the mitochondrial and cytosolic roGFP2-Orp1-based $H_2O_2$ sensors in ciPTC exposed to calcineurin inhibitors or hydrogen peroxide.** For experimental details, see legend to Fig 1. The percentages represent the mean 400/480 nm ratiometric response ratios compared to the averages of the corresponding non-treated control cells, which were set to 100%. Given that, under basal conditions, peroxisomal roGFP2-Orp1 is almost fully oxidized in ciPTC, this sensor was not included in this analysis. One-way ANOVA (OWA) test were used to calculate p-values for more than 2 groups. Paired t-test (PPT) and Wilcoxon test (WT) were used to calculate the difference between 2 groups. C, cytosolic roGFP2; ciPTC, conditionally immortalized proximal tubule cell; CsA, cyclosporin A; MT, mitochondrial roGFP2; Tac, tacrolimus.
(PDF)

**S1 References. References cited in S3 Table.**
(PDF)

## Acknowledgments

We thank Dr. Jean-Paul Decuypere (Department of Growth and Regeneration, KU Leuven) for kindly providing technical guidance and Ms. Inge Bongaers (Laboratory of Pediatric Nephrology, KU Leuven) for her technical assistance.

## Author Contributions

**Conceptualization:** Yasaman Ramazani, Dirk J. Kuypers, Elena Levtchenko, Lambert P. van den Heuvel, Marc Fransen.

**Data curation:** Yasaman Ramazani, Sante Princiero Berlingerio, Oyindamola Christiana Adebayo.

**Formal analysis:** Yasaman Ramazani, Sante Princiero Berlingerio, Celien Lismont, Lambert P. van den Heuvel, Marc Fransen.

**Funding acquisition:** Noël Knops, Elena Levtchenko, Marc Fransen.

**Investigation:** Lambert P. van den Heuvel, Marc Fransen.

**Methodology:** Yasaman Ramazani, Sante Princiero Berlingerio, Oyindamola Christiana Adebayo, Celien Lismont, Lambert P. van den Heuvel, Marc Fransen.

**Project administration:** Yasaman Ramazani.

**Resources:** Noël Knops, Dirk J. Kuypers, Elena Levtchenko, Marc Fransen.

**Software:** Yasaman Ramazani, Celien Lismont.

**Supervision:** Noël Knops, Lambert P. van den Heuvel, Marc Fransen.

**Validation:** Yasaman Ramazani.

**Visualization:** Yasaman Ramazani.

**Writing – original draft:** Yasaman Ramazani, Dirk J. Kuypers, Elena Levtchenko, Lambert P. van den Heuvel, Marc Fransen.

**Writing – review & editing:** Yasaman Ramazani, Noël Knops, Celien Lismont, Lambert P. van den Heuvel, Marc Fransen.

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
