## [Decision Letter · Decision Letter 0]

19 Jan 2021

PONE-D-20-36584

Therapeutic concentrations of calcineurin inhibitors do not deregulate glutathione redox balance in human renal proximal tubule cells

PLOS ONE

Dear Dr. Fransen,

Thank you for submitting your manuscript to PLOS ONE. After careful consideration, we feel that it has merit but does not fully meet PLOS ONE’s publication criteria as it currently stands. Therefore, we invite you to submit a revised version of the manuscript that addresses the points raised during the review process.

In particular, there is strong concern about whether by using a single technique the investigators can determine accurately changes in GSH balance and whether calcineurin is inducing oxidative stress that is not evidenced by changes in GSH (see both editor and reviewers comments below)

We look forward to receiving your revised manuscript.

Kind regards,

Rodrigo Franco

Academic Editor

PLOS ONE

Journal Requirements:

"Our journal requires that methods are described in enough detail to allow suitably skilled investigators to fully replicate your study. If materials, methods, and protocols are well established, authors may cite articles where those protocols are described in detail, but the submission should include sufficient information to be understood independent of these references. Please revise your manuscript so that cited protocols for Cell culture, RoGFP2 sensors and transfection, and Live-cell imaging/ quantitative image analysis are briefly but sufficiently described. For more information please see https://journals.plos.org/plosone/s/submission-guidelines#loc-materials-and-methods.

3. Please provide additional details regarding participant consent. In the ethics statement in the Methods and online submission information, please ensure that you have specified:

 - whether consent was obtained

 - whether consent was informed

 - what type of consent you obtained (for instance, written or verbal, and if verbal, how it was documented and witnessed).

 - if your study included minors, state whether you obtained consent from parents or guardians.

 - if the need for consent was waived by the ethics committee, please include this information.

Additional Editor Comments:

This manuscript reviews the effect of therapeutic concentrations of calcineurin on the glutathione redox balance of renal cells. The results are new but descriptive, requiring extensive experimental work to address the concerns of the reviewer (see comments below) which can be summarized as follows

1. Using only the roGFP2 sensor might not be enough to demonstrate that therapeutic concentrations of calcineurin do not affect renal GSH homeostasis. To begin with, roGFP should be preferentially coupled with glutaredoxin (grx) to increase its sensitivity and selectivity for changes in GSH. The investigators should perform additional experiments to corroborate that calcineurin does not alter overall GSH and redox balance: 1) using both biochemical (enzymatic assays) and fluorescence based reporters (mBcl) the investigators should corroborate their findings with roGFP at least within the cytosol; 2) the investigators should corroborate the responsiveness of the reporters to inhibition of GSH synthesis (BSO) or overall GSH depletion (NEM); 3) while calcineurin by itself might not induce an alteration in GSH/GSSG balance it is possible that an increase in GSH synthesis counteracts possible alterations in GSH/redox balance induced by calcineurin. Thus the investigators should evaluate if GSH depletion induced by BSO is enhanced by calcineurin

2. Calcineurin might be inducing a redox stress that is not reflected by GSH depletion. Thus the investigators need to demonstrate that calcineurin does not enhance the sensitivity of renal cells to oxidative stress and evaluate if calcineurin does not induce oxidative stress per se (evaluation of ROS formation in different compartments)

Reviewers' comments:

Reviewer's Responses to Questions

**Comments to the Author**

1. Is the manuscript technically sound, and do the data support the conclusions?

Reviewer #1: Partly

2. Has the statistical analysis been performed appropriately and rigorously? 

Reviewer #1: Yes

3. Have the authors made all data underlying the findings in their manuscript fully available?

Reviewer #1: Yes

4. Is the manuscript presented in an intelligible fashion and written in standard English?

Reviewer #1: Yes

5. Review Comments to the Author

Reviewer #1: The manuscript of Yasaman et al. aims to demonstrate the therapeutic concentrations of calcineurin inhibitors do not deregulate glutathione redox balance in human renal proximal tubule cells. A novel proximal tubule cell model established from human allograft biopsies that functionally expresses the relevant enzymes and transporters for the metabolism/efflux of calcineurin inhibitors and exposed these cells to therapeutic calcineurin inhibitor concentrations. The studies are well designed and in large part of the data support the conclusions of the authors. However, a few comments and questions are provided for clarification and some additional evidence is encouraged.

1. The authors should use additional method to measure glutathione redox balance to check whether endogenous glutathione level altered by calcineurin inhibitor.

2. It would be good to deplete glutathione (using BSO or others) and check whether calcineurin inhibitors induced any toxicity or synergize BSO toxicity to confirm calcineurin inhibitors does not deregulate glutathione redox balance in human renal proximal tubule cells

6. PLOS authors have the option to publish the peer review history of their article (what does this mean?). If published, this will include your full peer review and any attached files.

Reviewer #1: No

---

## [Author Response · Author response to Decision Letter 0]

2 Apr 2021

The authors thank the Reviewer and Editor for their thoughtful comments and constructive criticisms, which were very helpful to improve the quality of the manuscript.

Reviewer #1

Concern 1 – The reviewer states: “The authors should use additional method to measure glutathione redox balance to check whether endogenous glutathione level altered by calcineurin inhibitor.”

As requested by the reviewer, we have performed an additional method (in casu glutathione metabolomics) to investigate the potential effects of calcineurin inhibitors (CNI) on endogenous glutathione levels. In brief, the results of these experiments confirm and extend our conclusion that therapeutic concentrations of CNI treatment do not deregulate glutathione redox balance in human renal proximal tubule cells (Fig 2). For more details, we refer the reviewer to the section “Academic Editor, Concern 1, Metabolomic analysis of glutathione species”).

Concern 2 – The reviewer states: “It would be good to deplete glutathione (using BSO or others) and check whether calcineurin inhibitors induced any toxicity or synergize BSO toxicity to confirm calcineurin inhibitors does not deregulate glutathione redox balance in human renal proximal tubule cells.”

As suggested by the reviewer, we now include experiments in which we have studied the synergistic effects of BSO-mediated glutathione depletion and CNI on glutathione redox balance and cell toxicity. Importantly, BSO in itself did not cause any significant changes in the oxidation state of the cytosolic glutathione redox sensor (S2 Fig). In addition, neither treatment with BSO nor therapeutic or supratherapeutic doses of Tac, alone or in combination, reduced cell viability significantly (Fig 3A). In contrast, supra-physiological concentrations of CsA promoted cell death, both in control and glutathione-depleted cells (Fig 3B). In summary, these findings strongly indicate that therapeutic CNI concentrations do not cause redox stress in GSH-depleted ciPTC, at least not in the time window considered here. Note that these experiments were performed by Ms. Oyindamola Christiana Adebayo, who now also co-authors the manuscript.

Academic Editor

Concern 1 – The reviewer states: “Using only the roGFP2 sensor might not be enough to demonstrate that therapeutic concentrations of calcineurin do not affect renal GSH homeostasis. To begin with, roGFP should be preferentially coupled with glutaredoxin (grx) to increase its sensitivity and selectivity for changes in GSH. The investigators should perform additional experiments to corroborate that calcineurin does not alter overall GSH and redox balance: 1) using both biochemical (enzymatic assays) and fluorescence based reporters (mBcl) the investigators should corroborate their findings with roGFP at least within the cytosol; 2) the investigators should corroborate the responsiveness of the reporters to inhibition of GSH synthesis (BSO) or overall GSH depletion (NEM); 3) while calcineurin by itself might not induce an alteration in GSH/GSSG balance it is possible that an increase in GSH synthesis counteracts possible alterations in GSH/redox balance induced by calcineurin. Thus the investigators should evaluate if GSH depletion induced by BSO is enhanced by calcineurin.”

RoGFP2 versus Grx-roGFP2 – As pointed out by the Editor, some researchers do indeed use fusion constructs between roGFP2 and glutaredoxin (Grx) to promote rapid and specific equilibration between the dithiol-disulfide pairs of roGFP2 and the local glutathione pool. However, given that (i) others have demonstrated that, inside cells, roGFP2 by itself can already rapidly equilibrate with the local glutathione pool [Meyer et al., 2007], (ii) we have previously shown that all roGFP2 reporters used in this study can respond quickly and reversibly to local changes in redox state in response to transient oxidative insults and intermittent starvation [Ivashchenko et al., 2011], and (iii) in this study, the time window between CNI treatment and roGFP2 measurements is one or two days (this time window is sufficiently long to ensure equilibration between the roGFP2 sensors and the local glutathione redox couple), we are convinced that repeating the (time-consuming) roGFP2-based experiments with Grx-roGFP2 sensors would yield similar conclusions. Importantly, this hypothesis is further strengthened by two other facts: first, we have unpublished results documenting that roGFP2 and Grx-roGFP2 do indeed yield comparable results when these redox sensors are used side-by-side for other experimental conditions with a similar time window (for an example, see “Supplemental Figure for review purposes only”); and second, our roGFP2-based conclusions are in line with the newly incorporated glutathione metabolomics data (see Fig 2). In addition, we now elaborate on this concern in the Discussion (see revised manuscript, lines 366 to 377). We hope the reviewer can agree with our point of view.

Metabolomic analysis of glutathione species – To document in an independent way that (physiological) CNI concentrations do not alter the overall glutathione levels and/or redox balance, we carried out a series of metabolomics studies. Importantly, in agreement with our roGFP2 data (Fig 1), none of the tacrolimus (0.3 and 50 µg/ml; 48 hours) or cyclosporin A (15 µg/ml; 48 hours) treatments tested yielded significant alterations, neither in the GSH or GSSG levels nor in the GSSG/GSH ratios (Fig 2). However, treatment of ciPTCs with 100 µM buthionine sulfoximine (BSO), a potent glutathione synthesis inhibitor, resulted in a strong depletion of the intracellular glutathione levels within 48 h (Fig 2). Note that this finding is in agreement with multiple other studies (e.g., [Dooley et al., 2004; Jones et al., 2004; Funke et al., 2011; Kolossov et al., 2014]). These findings are now included and discussed in the revised version of the manuscript (see lines 232-242). Finally, given that others have reported that monochlorobimane (mBCL) is highly selective for glutathione in rodent cells, but – due to its low affinity for human glutathione S-transferases – also inefficient to visualize glutathione levels in human cells [Hedley and Chow, 1994], we did not use this reagent in our study.

Responsiveness of roGFP2 to BSO- and NEM-induced glutathione perturbations – As requested, we also investigated the responsiveness of cytosolic roGFP2 to BSO. Interestingly, BSO treatment did not cause significant changes in the oxidation state of the sensor (see S2 Fig). At first sight, this may be surprising. However, similar findings have been reported by others [Kolossov et al., 2014]. In addition, it can be expected that the relationship between GSH deficiency and subcellular oxidation may vary depending on cell type [Dooley et al., 2004; Funke et al., 2011; Ivashchenko et al., 2011], the subcellular localization of the sensor [Kolossov et al., 2014], and the dose and duration of BSO treatment [Jones et al., 2004]. This information is now also included in the Discussion (see revised manuscript, lines 242 to 250). Importantly, given that others have previously shown that NEM acts as a highly effective roGFP2-thiol-blocking agent that prevents sensor oxidation [Albrecht et al., 2011], it does not make sense to use this GSH-depleting agent to corroborate the validity of the roGFP2 sensors.

Synergistic effects of BSO and CNI on the intracellular GSH/GSSG balance – For more details, see “Reviewer 1, Concern 2”.

Concern 2 – The reviewer states: “Calcineurin might be inducing a redox stress that is not reflected by GSH depletion. Thus the investigators need to demonstrate that calcineurin does not enhance the sensitivity of renal cells to oxidative stress and evaluate if calcineurin does not induce oxidative stress per se (evaluation of ROS formation in different compartments).”

As already detailed above (see “Review 1, concern 2”), we now include data demonstrating that co-treatment of ciPTC with BSO and therapeutic CNI concentrations does not lead to alterations in the cytosolic glutathione redox state (S2 Fig). These new findings strongly indicate that CNI do not cause redox stress in GSH-depleted renal cells. This hypothesis is further strengthened by our observations that CNI concentrations within the therapeutic range do not alter the oxidation state of cytosolic and intra-mitochondrial roGFP2-ORP1, a highly responsive hydrogen peroxide sensor (see S3 Fig and S4 Table). As such, we hope that the Editor can agree with our point of view that therapeutic concentrations of calcineurin inhibitors do not deregulate the redox balance in human renal proximal tubule cells. 

Other editorial comments

Comment 1 – “Please ensure that your manuscript meets PLOS ONE's style requirements, including those for file naming. The PLOS ONE style templates can be found at: https://journals.plos.org/plosone/s/file?id=wjVg/PLOSOne_formatting_sample_main_body.pdf

and

We ensure that the manuscript and file naming meet all PLOS ONE style requirements. 

Comment 2 – “Our journal requires that methods are described in enough detail to allow suitably skilled investigators to fully replicate your study. If materials, methods, and protocols are well established, authors may cite articles where those protocols are described in detail, but the submission should include sufficient information to be understood independent of these references. Please revise your manuscript so that cited protocols for cell culture, roGFP2 sensors and transfection, and live-cell imaging/ quantitative image analysis are briefly but sufficiently described. For more information please see https://journals.plos.org/plosone/s/submission-guidelines#loc-materials-and-methods.”

We now elaborate in detail on how the ciPTC were cultured (see revised manuscript, lines 89 to 111) and transfected (see revised manuscript, lines 118 to 135). In addition, we now include sufficient details in the “Materials and Methods” section to allow suitably skilled investigators to fully replicate the roGFP2 imaging and image analysis protocols (see revised manuscript, lines 143 to 153). Importantly, note that the latter protocols have previously already been described in detail in a methods paper [Lismont et al., 2017]. Finally, we now also include detailed descriptions of the new procedures that were implemented to carry out the requested experiments (for the “Metabolomic analysis of glutathione species”, see revised manuscript, lines 155 to 178; for the “Cell viability assay”, see revised manuscript, lines 180 to 191).

Comment 3 – “Please provide additional details regarding participant consent. In the ethics statement in the Methods and online submission information, please ensure that you have specified (a) whether consent was obtained, (b) whether consent was informed, (c) what type of consent you obtained (for instance, written or verbal, and if verbal, how it was documented and witnessed), (d) if your study included minors, state whether you obtained consent from parents or guardians, and (e) if the need for consent was waived by the ethics committee, please include this information.”

The ethics statement was updated as follows: “The study protocol was approved by the Institutional Ethical Board (B32220109632 and B322201317188) and written consent was obtained from all study subjects or their parents. The study is registered with the clinical trial number of NCT01767350.” (see revised manuscript, lines 114 to 116).

Comment 4 – “We note that you have included the phrase “data not shown” in your manuscript. Unfortunately, this does not meet our data sharing requirements. PLOS does not permit references to inaccessible data. We require that authors provide all relevant data within the paper, Supporting Information files, or in an acceptable, public repository. Please add a citation to support this phrase or upload the data that corresponds with these findings to a stable repository (such as Figshare or Dryad) and provide and URLs, DOIs, or accession numbers that may be used to access these data. Or, if the data are not a core part of the research being presented in your study, we ask that you remove the phrase that refers to these data.”

As requested, we have removed the phrase “data not shown” and replaced this statement by “data submitted for publication” (authors: Knops N, Ramazani Y, De Loor H, Goldschmeding R, Nguyen TQ, van den Heuvel LP, Levtchenko E, Kuypers DJ; title: “Tacrolimus induces a pro-fibrotic response in donor-derived human proximal tubule cells dependent on common variants of CYP3A5 and ABCB1 genes”). We hope that this decision is acceptable for your journal.

---

## [Decision Letter · Decision Letter 1]

19 Apr 2021

Therapeutic concentrations of calcineurin inhibitors do not deregulate glutathione redox balance in human renal proximal tubule cells

PONE-D-20-36584R1

Dear Dr. Fransen,

We’re pleased to inform you that your manuscript has been judged scientifically suitable for publication and will be formally accepted for publication once it meets all outstanding technical requirements.

Kind regards,

Rodrigo Franco

Academic Editor

PLOS ONE

Additional Editor Comments (optional):

Reviewers' comments:

Reviewer's Responses to Questions

**Comments to the Author**

1. If the authors have adequately addressed your comments raised in a previous round of review and you feel that this manuscript is now acceptable for publication, you may indicate that here to bypass the “Comments to the Author” section, enter your conflict of interest statement in the “Confidential to Editor” section, and submit your "Accept" recommendation.

Reviewer #1: All comments have been addressed

2. Is the manuscript technically sound, and do the data support the conclusions?

Reviewer #1: Yes

3. Has the statistical analysis been performed appropriately and rigorously? 

Reviewer #1: Yes

4. Have the authors made all data underlying the findings in their manuscript fully available?

Reviewer #1: Yes

5. Is the manuscript presented in an intelligible fashion and written in standard English?

Reviewer #1: Yes

6. Review Comments to the Author

Reviewer #1: The authors addressed most of the concerns in a logical manner. I suggest to consider it for publication.

7. PLOS authors have the option to publish the peer review history of their article (what does this mean?). If published, this will include your full peer review and any attached files.

Reviewer #1: No

---

## [Editor Report · Acceptance letter]

21 Apr 2021

PONE-D-20-36584R1 

Therapeutic concentrations of calcineurin inhibitors do not deregulate glutathione redox balance in human renal proximal tubule cells 

Dear Dr. Fransen:

I'm pleased to inform you that your manuscript has been deemed suitable for publication in PLOS ONE. Congratulations! Your manuscript is now with our production department. 

Kind regards, 

on behalf of

Dr. Rodrigo Franco 

Academic Editor

PLOS ONE